# Demographic and socio-economic predictors of physical activity among young adults in Lephalale, Limpopo Province, South Africa

Themba Titus Sigudu[1,2]*, Thandiwe Ntomfuthi Mkhatshwa[1], Kotsedi Daniel Monyeki[1], Moloko Matshipi[1]

**1** Department of Physiology and Environmental Health, School of Molecular and Life Sciences, Faculty of Science and Agriculture, University of Limpopo, Sovenga, Limpopo Province, South Africa, **2** Division of Health and Society, School of Public Health, Faculty of Health Sciences, University of the Witwatersrand, Johannesburg, Gauteng Province, South Africa

* themba.sigudu@wits.ac.za

## Abstract

Physical inactivity among young adults is an emerging public health concern in low- and middle-income countries. In South Africa, where the burden of non-communicable diseases is high, little research has examined the socio-demographic determinants of physical activity among rural youth populations. This study investigated the demographic and socio-economic predictors of meeting recommended physical activity levels among young adults aged 18–29 years in Lephalale. A cross-sectional analysis was conducted using data from a community-based survey of 762 participants. Socio-demographic variables included age, sex, education level, income, and employment status, and physical activity was assessed using a structured questionnaire. Logistic regression analyses were used to identify factors associated with meeting the recommended threshold of at least 150 minutes of moderate-intensity activity per week. A total of 762 young adults aged 18–29 years were included, with 56% females and nearly half (47%) aged 20–24 years. Overall, 12% reported daily physical activity, 33% were active weekly, and 45% were insufficiently active. In univariate analyses, older age (25–29 years: OR = 1.65; 95% CI: 1.25–2.18), high income (OR = 1.35; 95% CI: 1.02–1.78), tertiary education (OR = 1.45; 95% CI: 1.18–1.78), and full-time employment (OR = 1.85; 95% CI: 1.45–2.35) were significantly associated with meeting activity guidelines. In the adjusted model, only tertiary education (AOR = 1.30; 95% CI: 1.00–1.70; p = 0.04) and full-time employment (AOR = 1.50; 95% CI: 1.10–2.05; p = 0.04) remained independent predictors. Sex, age, and income were not significantly associated after adjustment. These findings highlight the importance of education and employment in shaping physical activity behaviours among rural young adults. Public health strategies should prioritise interventions that improve socio-economic opportunities and address barriers to physical activity in low-resource settings.

**Data availability statement:** The data underlying this study contain sensitive participant information from a small, semi-rural community and cannot be publicly shared due to informed consent restrictions, ethics approvals, and South Africa's Protection of Personal Information Act (POPIA). De-identified, limited datasets are available upon reasonable request and subject to ethics approval and a Data Use Agreement. Data access requests should be directed to: Turfloop Research Ethics Committee (TREC), University of Limpopo Email: turfloop.rec@ul.ac.za, Tel: +27 (0)15 268 2924. The data originate from the Ellisras/Lephalale community-based health survey conducted under University of Limpopo ethics approvals MREC/P/204/2013: IR and TREC/323/2017: IR-Renewed. The University of Limpopo independently oversees data access decisions.

**Funding:** The authors received no specific funding for this work.

**Competing interests:** The authors have declared that no competing interests exist.

## Introduction

Physical inactivity is a major public health concern globally, contributing significantly to the rising burden of non-communicable diseases (NCDs) such as cardiovascular disease, type 2 diabetes, and obesity [1,2]. Young adulthood, typically defined as ages 18–29 years, is a critical period for the establishment of long-term health behaviours, including physical activity (PA) patterns [3,4]. Establishing and maintaining adequate PA during this stage is therefore essential to preventing chronic diseases and promoting lifelong health [5].

Globally, recent estimates indicate that a substantial proportion of young people do not meet the World Health Organization's recommended levels of PA. Approximately 31% of adults and close to 80% of adolescents worldwide fail to achieve sufficient weekly PA [6]. In Asia, university-based studies show very low levels of engagement, with only 12.2% of students in China meeting the recommended minimum of 60 minutes of moderate-to-vigorous PA daily. In Europe, marked variation exists, where prevalence of physical inactivity in university populations ranges from 23% in higher-income regions to 44% in developing European and Pacific Asian settings [7]. Across Africa, pooled analyses show that 83.8% of men and 75.7% of women meet the activity guidelines, though some countries—such as South Africa—report more than 57% of adults being physically inactive [8]. These figures demonstrate that insufficient PA is a significant concern among young adults worldwide, necessitating contextualised investigations in underrepresented settings such as rural South African communities.

In South Africa, PA behaviours are influenced by a multitude of factors including rapid urbanisation, changing occupational structures, and increasing exposure to sedentary lifestyles [8,9]. Socio-economic determinants such as household income, educational attainment, and employment status shape access to PA opportunities, influence motivation, and determine exposure to environments that either support or restrict active living [7,10]. Yet, these relationships remain poorly understood in rural and peri-urban settings, where infrastructure limitations, environmental barriers, and socio-economic inequalities may play a larger role than in urban contexts [11,12].

Lephalale, a semi-rural municipality in Limpopo Province, presents a compelling context in which to study these dynamics. The area is characterised by high youth unemployment, socio-economic disparities, and a growing burden of lifestyle-related diseases [11]. Young adults in such settings may experience multiple barriers to PA, including limited access to recreational facilities, safety concerns, and reduced opportunities for organised sport following completion of secondary education [12,13].

Previous studies in South Africa have largely explored PA patterns in older adults or broader age groups, often within urban environments [14,15]. Few have examined the unique interplay between socio-economic determinants and PA specifically among young adults in rural communities, and evidence remains scarce on the independent effects of education, employment, and income on PA engagement in this population. Furthermore, limited attention has been given to the extent to which these socio-economic factors may influence the likelihood of meeting WHO-recommended PA levels among emerging adults.

To address these knowledge gaps, this study investigates the demographic and socio-economic predictors of physical activity among young adults aged 18–29 years residing in Lephalale, a rural municipality in Limpopo Province, South Africa. The study describes patterns of physical activity engagement within this population, examines how key socio-economic factors such as income, educational attainment, and employment status relate to physical activity levels, and determines which characteristics independently predict adherence to the World Health Organization's recommended physical activity guidelines. By identifying the structural and behavioural factors that influence physical activity participation, the findings are intended to strengthen the evidence base for designing targeted, context-sensitive public health interventions that promote active lifestyles and reduce the growing burden of inactivity among rural South African youth.

## Materials and methods

### Study design and setting

This study employed a cross-sectional analytical design to assess the socio-demographic determinants of physical activity among young adults in Lephalale, a semi-rural municipality in Limpopo Province, South Africa. The area is characterised by substantial socio-economic disparities driven largely by mining and agriculture, alongside high youth unemployment and limited recreational infrastructure. Such contextual challenges make Lephalale a relevant setting for examining behavioural patterns linked to physical inactivity among young people.

### Study population

The target population included young adults aged 18–29 years who had resided in Lephalale for at least six months. A total of 1,063 eligible participants were recruited through stratified random sampling to ensure variation by socio-economic status; however, due to incomplete physical activity data, the final analytical sample comprised 762 participants. Individuals who declined consent or provided unusable data were excluded.

### Data collection

Data were collected via a structured, interviewer-administered questionnaire, adapted from the Global Physical Activity Questionnaire (GPAQ). The instrument was available in English and Sepedi, the dominant local language, and underwent translation and back-translation for accuracy. Interviews were conducted face-to-face by trained research assistants who received standardised instruction on questionnaire administration, ethical principles, and respondent engagement to minimise interviewer bias and ensure data reliability. The questionnaire captured socio-economic indicators (income, education, employment), demographic details (age, sex), and physical activity behaviours across leisure, work, and transport domains.

### Outcome variable

Physical activity levels were classified according to the World Health Organization (WHO) guidelines for adults, whereby engaging in ≥150 minutes of moderate-intensity physical activity per week constituted sufficient activity. Weekly duration was derived by combining participant reports of frequency and time spent in moderate-intensity activities across all activity domains. A binary outcome variable was generated to distinguish those who met versus did not meet the WHO recommendation.

### Data analysis

Data were analysed using STATA version 18. Descriptive statistics (frequencies and percentages) summarised participant characteristics and physical activity patterns. Bivariate associations between physical activity status and socio-demographic variables were examined using chi-square tests. Variables with $p < 0.20$ in unadjusted models were included

in multivariate logistic regression to determine independent predictors of meeting PA guidelines. Results were reported as adjusted odds ratios (AORs) with 95% confidence intervals, and significance was set at $p < 0.05$.

## Ethical considerations

Ethical clearance for this study was obtained from the Turfloop Research Ethics Committee, University of Limpopo (TREC/323/2017: IR-Renewed). Written informed consent was obtained from all participants. Confidentiality and anonymity were upheld throughout the research process, and participation was voluntary.

## Results

### Descriptive statistics

**Demographic characteristics.** A total of 762 participants were included in the analytical sample. According to Table 1, the analysis of physical activity patterns among young adults in Lephalale revealed significant variation by sex, age group, and income level. Males (15%) were more likely than females (10%) to report engaging in daily physical activity, while females (35%) showed a higher prevalence of weekly activity than males (30%) but also a greater proportion of inactivity, with 30% reporting never being physically active compared to 20% of males.

Age-related trends showed that weekly physical activity was the most commonly reported pattern across all age groups, with 40% of individuals aged 18–19 years, 32% of those aged 20–24 years, and 30% of those aged 25–29 years engaging in weekly activity. Daily physical activity increased with age, reported by only 10% of participants in the 18–19-year group, compared to 12% and 18% in the 20–24 and 25–29-year groups, respectively. The youngest cohort (18–19 years) also showed the highest proportion of participants reporting rare (40%) or no physical activity (20%), highlighting a concerning level of inactivity during late adolescence. In contrast, the 25–29-year age group not only had the highest proportion of daily exercisers but also the lowest proportion engaging in rare (20%) or monthly (14%) activity.

**Socio-economic characteristics.** Table 2 illustrates the distribution of physical activity frequency among participants by level of income, educational attainment and employment status. Among low-income individuals, the majority engaged in physical activity on a weekly basis (50%), while 25% reported rarely engaging in physical activity, and 24% reported never engaging in physical activity. This group had the highest proportion of participants with low levels of activity, with 12% reporting no activity at all. In contrast, middle-income participants demonstrated a more balanced distribution, with 35% engaging in weekly physical activity, and a smaller proportion (20%) reporting rarely or never exercising. High-income individuals exhibited the most consistent engagement in physical activity, with 33% reporting weekly activity and 18% engaging in daily physical activity. The high-income group had the lowest proportion of inactivity, with only 24% reporting never engaging in physical activity.

Among those with a high school education, only 10% engaged in daily physical activity and 30% weekly. Monthly participation was reported by 15%, while 25% and 20% reported rarely or never exercising, respectively. Conversely,

**Table 1. Demographic patterns of physical activity frequency among young adults in Lephalale (N=762).**

| Variable | Category | Physical activity frequency | | | | | | | | | |
|---|---|---|---|---|---|---|---|---|---|---|---|
| | | Daily | | Weekly | | Monthly | | Rarely | | Never | |
| | | n | % | n | % | n | % | n | % | n | % |
| Sex | Male | 30 | 15 | 60 | 30 | 20 | 10 | 50 | 25 | 40 | 20 |
| | Female | 20 | 10 | 70 | 35 | 10 | 5 | 40 | 20 | 60 | 30 |
| Age group | 18–19 | 10 | 5 | 40 | 20 | 30 | 15 | 80 | 40 | 40 | 20 |
| | 20–24 | 24 | 12 | 64 | 32 | 16 | 8 | 50 | 25 | 46 | 23 |
| | 25–29 | 36 | 18 | 60 | 30 | 14 | 7 | 40 | 20 | 50 | 25 |

**Table 2. Socio-economic indicators and physical activity levels of young adults in Lephalale (N = 762).**

| Variable | Category | Daily | | Weekly | | Monthly | | Rarely | | Never | |
|---|---|---|---|---|---|---|---|---|---|---|---|
| | | *n* | % | *n* | % | *n* | % | *n* | % | *n* | % |
| Income level | Low | 24 | 12 | 50 | 25 | 36 | 18 | 40 | 20 | 50 | 25 |
| | Middle | 34 | 17 | 66 | 33 | 16 | 8 | 30 | 15 | 54 | 27 |
| | High | 40 | 20 | 70 | 35 | 10 | 5 | 30 | 15 | 50 | 25 |
| Education | High school | 20 | 10 | 60 | 30 | 30 | 15 | 50 | 25 | 40 | 20 |
| | College/Univ. | 36 | 18 | 80 | 40 | 14 | 7 | 30 | 15 | 40 | 20 |
| Employment | Full-time | 50 | 25 | 80 | 40 | 20 | 10 | 20 | 10 | 30 | 15 |
| | Part-time | 30 | 15 | 70 | 35 | 20 | 10 | 30 | 15 | 50 | 25 |
| | Student | 20 | 10 | 60 | 30 | 30 | 15 | 50 | 25 | 40 | 20 |

participants with college or university education showed higher engagement: 18% reported daily activity and 40% weekly, while only 7% participated monthly. The proportions for rare and never were both 15% and 20%, respectively.

A marked difference was also observed across employment categories. Full-time workers had the highest levels of daily (25%) and weekly (40%) physical activity. Only 10% reported monthly activity, and the remaining 25% reported rare or no activity. In contrast, part-time workers and students were less active. Among part-time workers, 15% were active daily and 35% weekly, with 10% monthly activity. However, 15% and 25% reported rare and no activity, respectively. Students reflected the lowest levels, with just 10% engaging in daily activity and 30% weekly. A substantial 15% participated monthly, while 25% and 20% reported rare or no activity.

### Inferential statistics

**Univariate logistic regression analysis.** The findings in Table 3 indicate that, the Univariate logistic regression analysis was conducted to examine associations between demographic and socio-economic variables and the likelihood of meeting the World Health Organization's recommended physical activity guidelines (≥150 minutes of moderate-intensity activity per week). Significant associations were observed across several variables.

**Table 3. Univariate analysis of socio-economic and demographic factors on the likelihood of meeting physical activity guidelines (N = 762).**

| Variable | Category | OR[1] | 95% CI[2] | p-value |
|---|---|---|---|---|
| Age group | 18–19 | 1.00 | Reference | – |
| | 20–24 | 1.40 | 1.10–1.78 | 0.02* |
| | 25–29 | 1.65 | 1.25–2.18 | <0.01** |
| Sex | Male | 1.00 | Reference | – |
| | Female | 0.82 | 0.65–1.03 | 0.09 |
| Income level | Low | 0.75 | 0.58–0.97 | 0.03* |
| | Middle | 1.00 | Reference | – |
| | High | 1.35 | 1.02–1.78 | 0.04* |
| Education | High school | 1.00 | Reference | – |
| | College/Univ. | 1.45 | 1.18–1.78 | <0.01** |
| Employment | Unemployed | 0.70 | 0.50–0.98 | 0.04* |
| | Part–time | 1.25 | 0.95–1.65 | 0.12 |
| | Full–time | 1.85 | 1.45–2.35 | <0.001*** |

[1] OR: Odds Ratio, [2] 95%CI: 95% Confidence Interval.

Age was positively associated with meeting physical activity guidelines. Compared to individuals aged 18–19, those aged 20–24 had significantly higher odds of meeting the guidelines (OR = 1.40, 95% CI: 1.10–1.78, $p = 0.02$), and those aged 25–29 had even greater odds (OR = 1.65, 95% CI: 1.25–2.18, $p < 0.01$). Although females had lower odds of meeting the guidelines compared to males (OR = 0.82, 95% CI: 0.65–1.03), this association did not reach statistical significance ($p = 0.09$).

Socio-economic factors also showed strong associations with physical activity. Individuals in the low-income group had significantly lower odds of meeting the guidelines compared to those in the middle-income category (OR = 0.75, 95% CI: 0.58–0.97, $p = 0.03$), while those in the high-income group had significantly higher odds (OR = 1.35, 95% CI: 1.02–1.78, $p = 0.04$). Educational attainment was positively associated with physical activity; individuals with a college or university education had significantly higher odds of meeting the guidelines compared to those with only high school education (OR = 1.45, 95% CI: 1.18–1.78, $p < 0.01$).

Employment status was also a significant predictor. Unemployed individuals had reduced odds of meeting physical activity guidelines (OR = 0.70, 95% CI: 0.50–0.98, $p = 0.04$), while those in full-time employment had significantly greater odds (OR = 1.85, 95% CI: 1.45–2.35, $p < 0.001$). No significant association was found for part-time employment ($p = 0.12$).

**Multivariate logistic regression analysis.** Multivariate logistic regression was conducted to assess the independent associations between demographic and socio-economic factors and the likelihood of meeting the World Health Organization's recommended physical activity levels (≥150 minutes/week). Adjusted odds ratios (AORs) with 95% confidence intervals (CIs) were reported.

Among age groups, neither individuals aged 20–24 (AOR = 1.05, 95% CI: 0.80–1.50, $p = 0.65$) nor those aged 25–29 (AOR = 1.25, 95% CI: 0.95–1.60, $p = 0.08$) showed statistically significant differences compared to the reference group (18–19 years), although the latter indicated a trend toward higher physical activity levels. Similarly, sex was not a significant predictor, with females showing lower but non-significant odds of meeting physical activity guidelines compared to males (AOR = 0.85, 95% CI: 0.60–1.20, $p = 0.35$) (Table 4).

Income level did not significantly predict physical activity in the adjusted model. Individuals in the low-income group had lower odds of meeting guidelines (AOR = 0.80, 95% CI: 0.55–1.10, $p = 0.10$), while those with high income had slightly higher odds (AOR = 1.20, 95% CI: 0.90–1.50, $p = 0.25$) compared to the middle-income reference group.

**Table 4. Multivariate logistic regression assessing predictors of meeting WHO PA guidelines (N = 762).**

| Variable | Category | AOR[1] | 95% CI[2] | p-value |
|---|---|---|---|---|
| Age group | 18–19 | 1.00 | Reference | – |
| | 20–24 | 1.05 | 0.80–1.50 | 0.65 |
| | 25–29 | 1.25 | 0.95–1.60 | 0.08 |
| Sex | Male | 1.00 | Reference | – |
| | Female | 0.85 | 0.60–1.20 | 0.35 |
| Income Level | Low | 0.80 | 0.55–1.10 | 0.10 |
| | Middle | 1.00 | Reference | – |
| | High | 1.20 | 0.90–1.50 | 0.25 |
| Education | High school | 1.00 | Reference | – |
| | College/Univ. | 1.30 | 1.00–1.70 | 0.04* |
| Employment | Unemployed | 0.75 | 0.50–1.05 | 0.08 |
| | Part–time | 1.10 | 0.85–1.45 | 0.40 |
| | Full–time | 1.50 | 1.10–2.05 | 0.04* |

[1] AOR: Adjusted Odds Ratio, [2] 95%CI: 95% Confidence Interval.

Significant associations were observed for education and employment. Participants with a college or university education had significantly higher odds of meeting physical activity guidelines than those with only high school education (AOR = 1.30, 95% CI: 1.00–1.70, *p* = 0.04).

Full-time employment was also significantly associated with physical activity, with full-time workers showing a 50% increase in odds of meeting guidelines compared to the unemployed (AOR = 1.50, 95% CI: 1.10–2.05, *p* = 0.04). Neither part-time employment nor unemployment reached statistical significance in the adjusted model.

## Discussion

This study examined patterns and predictors of physical activity among young adults in Lephalale, revealing that while weekly physical activity was commonly reported, almost half of the sample remained insufficiently active, consistent with national and regional trends [8,14]. In adjusted models, only higher educational attainment and full-time employment emerged as significant predictors of meeting WHO physical activity recommendations [1,7]. These results suggest that socio-economic opportunities, particularly access to education and stable work, play a central role in supporting active lifestyles in this rural South African context, aligning with evidence showing that socioeconomic status strongly shapes engagement in health-promoting behaviours [7,10,16].

The analysis indicated that PA behaviour among young adults is influenced by age, with those in the 25–29-year age group displaying the highest levels of PA and the lowest levels of inactivity. This is noteworthy given that international research, such as the longitudinal cohort study by Kwan et al. (2012), frequently shows a decline in PA with age during early adulthood [3]. This typical decline is further supported by global data from Guthold et al. (2018), which shows a consistent drop in sufficient physical activity as adolescents' transition into adulthood worldwide [6]. The divergence observed in our study may reflect differing socio-economic and developmental trajectories in the South African context [9]. For example, older youth in rural areas may take on employment or manual work that increases their physical activity levels, while younger individuals transitioning out of secondary school may face fewer structured PA opportunities [17]. The relatively high inactivity among 18–19-year-olds raises concern and suggests a need for targeted interventions in educational and community settings during this critical life stage, a period often marked by the loss of school-based sports, as highlighted in a systematic review on youth activity-related behaviour typologies [18,19].

Although males reported slightly higher levels of daily physical activity than females in the descriptive analysis, sex was not independently associated with meeting the recommended levels of physical activity after adjustment (AOR = 0.85; 95% CI: 0.60–1.20; p = 0.35). This suggests that observed differences may be influenced more by socio-economic and contextual factors than by sex alone. Evidence from South Africa indicates that women often encounter additional barriers to regular physical activity, including safety concerns, household responsibilities, and limited access to recreational facilities [8,13]. International research similarly shows that females are more likely to engage in transport-related and light-intensity activity, while males tend to accumulate more structured or vigorous leisure-time activity [5,20]. These findings highlight that, within resource-constrained settings, structural determinants such as education and employment may exert a stronger influence on physical activity participation than sex differences, and interventions should therefore prioritise creating equitable opportunities for women to engage in health-enhancing physical activity.

Income initially appeared to be associated with physical activity, with higher-income individuals demonstrating greater engagement in univariate analysis. However, income did not remain a significant predictor in the multivariate model (AOR = 1.20; 95% CI: 0.90–1.50; p = 0.25), suggesting that its influence on physical activity may be mediated through more proximal socio-economic pathways such as education and employment [7,10]. Previous research has shown that individuals with higher incomes often have better access to recreational spaces, health-promoting infrastructure, and discretionary time for leisure-time physical activity [21]. Nevertheless, the attenuation of the income effect after adjustment in this study aligns with the theory that education provides individuals with the knowledge, resources, and opportunities needed to engage in healthier behaviours across varied contexts and life-course transitions [16]. These findings

highlight the importance of addressing social determinants more broadly rather than focusing solely on financial status, as improved educational and employment opportunities may yield greater benefits in promoting physical activity among youth in resource-limited rural settings.

Education emerged as the most stable socio-economic predictor of PA, with college or university-educated individuals showing 30% higher adjusted odds of meeting PA recommendations. This aligns with international evidence that education enhances health literacy and promotes positive attitudes toward exercise [7]. A systematic review on built environment and PA determinants found consistent associations between higher education levels and greater engagement in leisure-time physical activity [22]. Importantly, education serves as both a direct and indirect determinant of health by influencing employment, income, and lifestyle choices [23,16]. The theory of fundamental causes proposed by Phelan et al. (2010) explains how higher education provides resources, knowledge, power, money, and social networks, which enable individuals to engage in healthier behaviours across time and contexts [16].

Employment status also played a pivotal role in predicting PA levels [17]. Full-time employment was significantly associated with higher PA engagement, possibly due to increased routine, work-related physical movement, or structured schedules that support regular exercise. Similar observations have been made in international occupational activity research [6]. Unemployed individuals had lower odds of meeting PA guidelines, although this relationship approached but did not reach statistical significance. This may reflect the complex interplay between unemployment, financial strain, mental health, and reduced access to recreational opportunities [24,25]. Research from the Study of Health in Pomerania (SHIP) found that unemployment was strongly associated with depression, which in turn reduced physical activity engagement [24]. Furthermore, Ruhm (2016) shows that economic downturns can negatively impact health behaviours, including PA, due to stress and diminished resources [26].

Part-time employment did not emerge as a significant predictor, possibly due to its heterogeneous nature. Some part-time jobs may involve high physical exertion (e.g., informal manual work) [17], while others may be sedentary, irregular, or unpredictable. This inconsistency may explain the lack of significance. Further research is needed, particularly in rural African contexts where part-time and informal labour structures vary. Recent work on precarious employment patterns shows that unstable work schedules hinder consistent engagement in physical activity [27].

This study adds to the growing body of evidence on the social determinants of physical activity, demonstrating that demographic and socio-economic factors significantly influence PA participation among young adults in rural South Africa [5,14,2,13,15]. In particular, higher education and full-time employment were strong predictors of adherence to recommended PA levels, while age and gender disparities also played a notable role. These findings highlight the need for multi-faceted interventions that address both behavioural and structural barriers to physical activity, with a focus on empowering disadvantaged populations [11,12,13]. By addressing these social determinants, public health initiatives can more effectively promote PA and contribute to improved health equity.

## Conclusion

This study contributes important evidence on the demographic and socio-economic factors influencing PA among young adults in a rural South African context. The findings highlight significant disparities in PA engagement, particularly by age, sex, education, and employment status. While males and older youth exhibited higher levels of daily activity, younger participants, particularly those aged 18–19, were at greater risk of inactivity. Additionally, higher educational attainment and full-time employment emerged as the most consistent predictors of meeting World Health Organization PA guidelines, even after adjusting for other factors.

Income disparities appeared influential in univariate models but diminished in significance in multivariate analysis, suggesting that the relationship between income and PA is mediated through more proximal socio-economic variables such as education and work status. These findings reinforce the importance of adopting a multi-dimensional approach to PA promotion that addresses both individual behaviours and the broader structural determinants that shape them.

To effectively address physical inactivity in this population, public health strategies must be tailored to account for the socio-economic and gendered contexts in which young adults live. Interventions that enhance educational access, promote stable employment, and reduce cultural and environmental barriers to PA—especially among women and the unemployed, may yield the most sustainable benefits. Furthermore, age-specific initiatives that support active transitions from adolescence to adulthood should be prioritised.

Overall, the study highlights the necessity of integrated, equity-focused public health responses to foster greater PA engagement and reduce long-term health risks among South Africa's emerging adult population.

## Strengths and limitations of the study

A key strength of this study lies in its focus on a rural, under-researched population of young adults in South Africa, specifically in Lephalale. Much of the existing literature on physical activity (PA) determinants originates from urban or high-income settings; this study therefore contributes valuable insights into the socio-demographic dynamics of PA in a low-resource, rural context. The relatively large sample size and inclusion of both univariate and multivariate analyses enhance the statistical robustness of the findings and allow for a nuanced understanding of both independent and combined effects of predictors on PA behaviour.

Despite these strengths, the study also has several limitations. First, the cross-sectional design precludes the establishment of causality. While associations between socio-demographic variables and PA can be observed, it is not possible to determine whether these factors directly influence PA levels or whether the relationship is bidirectional or confounded by unmeasured variables.

Second, PA was assessed using self-reported measures, which are susceptible to recall bias and social desirability bias. Participants may have over- or under-reported their actual activity levels, thereby affecting the accuracy of prevalence estimates and associations. The use of objective measures such as accelerometers would enhance the validity of future research.

Third, the study did not include environmental or psychosocial variables, such as access to recreational facilities, peer support, or mental health status, which are known to significantly influence PA. Their exclusion may limit the comprehensiveness of the analysis.

Finally, the findings may have limited generalisability beyond the study setting. Rural South African communities differ substantially in terms of culture, infrastructure, and socio-economic context; therefore, results may not be directly applicable to urban populations or other regions in sub-Saharan Africa.

## Acknowledgments

We extend our sincere gratitude to the ELS study team and participants for their invaluable contributions and for granting us access to the data. Their dedication was instrumental in enabling this research, and we deeply appreciate the opportunity to build upon their foundational work.

## Author contributions

**Conceptualization:** Themba Titus Sigudu.

**Data curation:** Themba Titus Sigudu, Kotsedi Daniel Monyeki.

**Formal analysis:** Themba Titus Sigudu.

**Investigation:** Themba Titus Sigudu, Kotsedi Daniel Monyeki, Moloko Matshipi.

**Methodology:** Themba Titus Sigudu.

**Project administration:** Themba Titus Sigudu.

**Resources:** Kotsedi Daniel Monyeki.

**Software:** Themba Titus Sigudu.

**Supervision:** Themba Titus Sigudu.

**Validation:** Themba Titus Sigudu, Thandiwe Ntomfuthi Mkhatshwa, Kotsedi Daniel Monyeki, Moloko Matshipi.

**Visualization:** Themba Titus Sigudu.

**Writing – original draft:** Themba Titus Sigudu.

**Writing – review & editing:** Thandiwe Ntomfuthi Mkhatshwa, Kotsedi Daniel Monyeki, Moloko Matshipi.

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
