## [Decision Letter · Decision Letter 0]

8 Aug 2025

PGPH-D-25-01139

Exploring the interplay between socio-economic and demographic factors in shaping physical activity behaviours among young adults in Elisrus

Dear Dr. Sigudu,

Thank you for submitting your manuscript to PLOS Global Public Health. After careful consideration, we feel that it has merit but does not fully meet PLOS Global Public Health’s publication criteria as it currently stands. Therefore, we invite you to submit a revised version of the manuscript that addresses the points raised during the review process.

Please note that we have only been able to secure a single reviewer to assess your manuscript. We are issuing a decision on your manuscript at this point to prevent further delays in the evaluation of your manuscript. Please be aware that the editor who handles your revised manuscript might find it necessary to invite additional reviewers to assess this work once the revised manuscript is submitted. However, we will aim to proceed on the basis of this single review if possible.

We look forward to receiving your revised manuscript.

Kind regards,

Jennifer Tucker, PhD

Staff Editor

Journal Requirements:

Additional Editor Comments (if provided):

Reviewers' comments:

Reviewer's Responses to Questions

**Comments to the Author**

1. Does this manuscript meet PLOS Global Public Health’s publication criteria?

Reviewer #1: Partly

2. Has the statistical analysis been performed appropriately and rigorously?

Reviewer #1: Yes

3. Have the authors made all data underlying the findings in their manuscript fully available (please refer to the Data Availability Statement at the start of the manuscript PDF file)?

Reviewer #1: No

4. Is the manuscript presented in an intelligible fashion and written in standard English?

Reviewer #1: No

Reviewer #1: 1. In paragraph 1 of the introduction, The (Kwan....) citation should be deleted as it is not following the right guideline of the journal citation preferences.

2. In the last paragraph of the introduction, where is Lephalale? the province, the country should be added

3. In the last paragraph of the introduction, With this specific aims highlighted here, it does not fully reflect in the tittle of the study. The PA pattern and intensity does not align well with the title. If these are aims, the title need to be changed or rephrased.

4. In Study design and setting, This seems jumping here and there when it comes to determining the main objective/purpose of this study. This should be adequately looked into.

5.Under the study population, red to black ink should be adopted. should black and white color.

6. Under the data collection, I am concerned about this, Since the survey used in this study was adapted from previous survey, what is the psychometric property of the newly adopted questionnaire? its reliability and validity of the new survey should be added. This could help those who wish to replicate the study elsewhere.

7. Under the ethical clearance, does this study have ethical numbers? this should be added.

8. Under the discussion in the first paragraph, Please, see my comment about the aim of the study above.

9. I suggest the paper be read by experts in the field who is qualified to judge the scientific writing as I noticed that the work need to be rigorously edited for clarity.

10. The author need to make sure that the study follow STROBE checklist as it does not do in the current status of the paper.

**Do you want your identity to be public for this peer review?** For information about this choice, including consent withdrawal, please see our Privacy Policy

Reviewer #1: **Yes:** Sunday O Onagbiye

---

## [Decision Letter · Decision Letter 1]

23 Oct 2025

PGPH-D-25-01139R1

Exploring the interplay between socio-economic and demographic factors in shaping physical activity behaviours among young adults in Elisrus

Dear Dr. Sigudu,

Thank you for submitting your manuscript to PLOS Global Public Health. After careful consideration, we feel that it has merit but does not fully meet PLOS Global Public Health’s publication criteria as it currently stands. Therefore, we invite you to submit a revised version of the manuscript that addresses the points raised during the review process.

We look forward to receiving your revised manuscript.

Kind regards,

Madhur Verma

Academic Editor

Journal Requirements:

Additional Editor Comments (if provided):

Reviewers' comments:

Reviewer's Responses to Questions

**Comments to the Author**

Reviewer #1: (No Response)

Reviewer #2: (No Response)

publication criteria?

Reviewer #1: Yes

Reviewer #2: Partly

3. Has the statistical analysis been performed appropriately and rigorously?

Reviewer #1: Yes

Reviewer #2: Yes

4. Have the authors made all data underlying the findings in their manuscript fully available (please refer to the Data Availability Statement at the start of the manuscript PDF file)?

Reviewer #1: Yes

Reviewer #2: No

5. Is the manuscript presented in an intelligible fashion and written in standard English?

Reviewer #1: No

Reviewer #2: Yes

Reviewer #1: The author should rephrase the title of the manuscript to read;

"Demographic and socio-economic predictors of PA among young adults in Elisrus"

Reviewer #2: Reviewing the manuscript, I have some questions and comments that authors need to further address and clarify throughout the manuscript before it could be considered for publication PLOS Global Public Health journal. Here are the questions, comments and clarifications throughout the manuscript that could be addressed or revised to improve the clarity and content of the manuscript. Attachment is enclosed herewith.

**Do you want your identity to be public for this peer review?** For information about this choice, including consent withdrawal, please see our Privacy Policy

Reviewer #1: **Yes:** Sunday O Onagbiye

Reviewer #2: **Yes:** Heng Sopheab

---

## [Decision Letter · Decision Letter 2]

1 Feb 2026

Demographic and socio-economic predictors of Physical Activity  among young adults in Lephalale, Limpopo Province, South Africa

PGPH-D-25-01139R2

Dear Dr Sigudu,

We are pleased to inform you that your manuscript 'Demographic and socio-economic predictors of Physical Activity  among young adults in Lephalale, Limpopo Province, South Africa' has been provisionally accepted for publication in PLOS Global Public Health.

Best regards,

Madhur Verma

Academic Editor

Reviewer Comments (if any, and for reference):

Reviewer's Responses to Questions

**Comments to the Author**

Reviewer #1: All comments have been addressed

Reviewer #2: All comments have been addressed

publication criteria?

Reviewer #1: Yes

Reviewer #2: Yes

3. Has the statistical analysis been performed appropriately and rigorously?

Reviewer #1: Yes

Reviewer #2: Yes

4. Have the authors made all data underlying the findings in their manuscript fully available (please refer to the Data Availability Statement at the start of the manuscript PDF file)?

Reviewer #1: Yes

Reviewer #2: No

5. Is the manuscript presented in an intelligible fashion and written in standard English?

Reviewer #1: Yes

Reviewer #2: Yes

Reviewer #1: NA

Reviewer #2: The authors have tried to respond fairly to all comments raised in the original manuscript in terms of unclarity, citing additional literature, inconsistency, overstatement of the findings. The revised manuscript has been improved significantly for its clarity, conciseness, consistency, and reading. Alot of efforts have been put in this revised version. I have no more comment.

**Do you want your identity to be public for this peer review?** For information about this choice, including consent withdrawal, please see our Privacy Policy

Reviewer #1: **Yes:** Sunday O. Onagbiye

Reviewer #2: No
